# Amphiphilic Bowl-Shaped Janus Particles Prepared via Thiol–Ene Click Reaction for Effective Oil–Water Separation

**DOI:** 10.3390/nano13030455

**Published:** 2023-01-22

**Authors:** Xian Qi, Yaxian Du, Ziqiang Zhang, Xu Zhang

**Affiliations:** National-Local Joint Engineering Laboratory for Energy Conservation of Chemical Process Integration and Resources Utilization, School of Chemical Engineering and Technology, Hebei University of Technology, Tianjin 300401, China

**Keywords:** Janus particles, bowl-shaped particles, thiol–ene click reaction, oil–water separation

## Abstract

Janus particles for oil–water separation have attracted widespread attention in recent years. Herein, we prepared a bowl-shaped Janus particle that could rapidly separate oil and water through a thiol–ene click reaction and selective etching. Firstly, snowman-like composite microspheres based on silica and mercaptopropyl polysilsesquioxane (SiO_2_@MPSQ) were prepared by a hydrolytic condensation reaction and phase separation, and the effects of the rotational speed and molar ratios on their microscopic morphologies were investigated. Subsequently, bowl-shaped Janus particles with convex hydrophilic and concave oleophilic surfaces were prepared via a thiol–ene click reaction followed by HF etching. Our amphiphilic bowl-shaped Janus particles could remarkably separate micro-sized oil droplets from an n-heptane–water emulsion with a separation efficiency of >98% within 300 s. Based on the experimental and theoretical results, we proposed the underlying mechanism for the coalescence of oil droplets upon the addition of the amphiphilic bowl-shaped Janus particles.

## 1. Introduction

As the requirements for sustainable development increase, oil–water separation has attracted global attention for oil removal from water and wastewater [1]. Deep-sea oil transportation accidents and fuel-oil spills have led to a large amount of oil entering the marine environment, causing serious damage to the ecosystem [2,3,4]. In general, oils are divided into floating and dispersed oils, as well as surfactant-stabilized and surfactant-free micro-sized oil droplets [4]. Considerable endeavors have been devoted to effectively separating floating and dispersed oils by controlling the surface wettability of superwettable materials [5,6,7,8,9]. Recently, superhydrophobic ceramic membrane and nanosheet materials have become new types of materials that can separate slick oil and stable oil droplets [10,11,12]. However, an effective surfactant-free separation method for micro-fine oil droplets has not been achieved thus far. Although traditional separation methods, such as gravity separation, mechanical separation, and biological methods are effective strategies [13,14,15,16,17], the separation of small oil droplets is still challenging because of the high costs and energy consumption, as well as low the separation efficiency [18].

Janus particles hold a promising opportunity to effectively separate oil from water because of their unique structural and physical properties. The word “Janus” comes from Roman mythology and refers to a double-faced god looking in opposite directions, one facing the future and the other facing the past [19]. Janus particles combine different properties, offering a wide range of applications such as surfactants, sensors, drug delivery, coatings, self-propelled carriers, and probes [20]. The morphology of Janus particles is critical to their performance. Thus far, various Janus particles have been prepared, exhibiting various shapes such as snowman [21,22,23,24], dumbbell [25], hamburger [26], and mushroom [27,28,29] shapes, among others. Hou et al. designed and prepared amphiphilic Janus particles, which can effectively emulsify various oils in seawater and overcome several drawbacks of conventional chemical dispersants [30]. Bradley et al. used a new one-step synthesis method to prepare amphiphilic polymer-based Janus particles with a highly uniform and adjustable wettability that can be used for selective post-functionalization reactions and as strong solid surfactants to stabilize heterogeneous air/water interfaces [31]. In addition, bowl-shaped particles have been the focus of some research because they can provide a concave structure with an inner cavity and a high specific surface area [32,33]. Their lipophilic surface can effectively capture micro-scaled oil droplets from water. 

In this study, we present a simple approach for the preparation of snowman-like composite particles. The snowman-like morphology could be systematically adjusted by controlling the monomer ratio, stirring rate, and template particle size. In addition, based on the snowman-like particles, amphiphilic bowl-shaped Janus particles were prepared by a thiol–ene click reaction and selective etching for the effective separation of oil droplets from water. The synthesis procedures of the snowman-like and bowl-shaped Janus particles (Figure 1), as well as the adsorption mechanism of the Janus particles at the oil–water interface, were investigated. This study presents a new direction for the preparation of bowl-shaped Janus particles and promotes their use for wastewater treatment.

## 2. Experimental Section

### 2.1. Materials

Tetraethyl orthosilicate (TEOS, 98%) and styrene (St, >99.5%) were purchased from Tianjin Chemical Reagent Co., Ltd., Tianjin, China, and purified by reduced-pressure distillation at an elevated temperature. An ammonia solution (NH_3_·H_2_O, 28%), acetonitrile (99%), anhydrous methanol (99%), ethanol, cetyltrimethylammonium bromide (CTAB, 99%), 3-mercaptopropyltrimethoxysilane (MPTMS, 95%), and acrylic acid (AA, 99%) were purchased from Aladdin Co., Ltd., Shanghai, China, and used without further purification. Further, 2,2-dimethoxy-2-phenylacetophenone (DMPA, 98%) was supplied by JiuDing Chemical Reagent Co., Ltd., Shanghai, China. Hydrofluoric acid (HF, 40%) was purchased from RHAWN Chemical Reagent Co., Ltd., Shanghai, China. Ultrapure water (resistivity > 18.2 MΩ·cm) was obtained using a Milli-Q reverse osmosis water purification system (Sichuan ULUPURE Ultrapure Technology Co., Ltd., Chengdu, China).

### 2.2. Synthesis of Snowman-like Composite Particles Based on Silica and Mercaptopropyl Polysilsesquioxane (SiO_2_@MPSQ) 

A typical procedure for synthesizing snowman-like composite particles is as follows. First, silica microspheres with different diameters were prepared according to the Stöber method [34]. Second, 0.1 g of dried silica microsphere powder was dispersed in 50 mL of ethanol by ultrasonication and further mixed uniformly with 25 mL of deionized (DI) water solution containing 0.02 g of CTAB. After 0.4 mL of ammonia solution was added to the mixture and stirred for 30 min, 250 μL of MPTMS was added to the solution. The reaction was performed by stirring at 38 °C for 5 h. SiO_2_@MPSQ snowman-like composite particles were obtained after centrifugation and washing twice with ethanol.

### 2.3. Preparation of Bowl-Shaped Janus Composite Particles (AA-MPSQ-St)

First, SiO_2_@MPSQ-AA snowman-like composite particles were prepared by a thiol–ene click reaction, and the typical procedure is described as follows. Snowman-like composite particles (0.1 g) were dissolved in 50 mL of ethanol. Then, 0.04 g of DMPA and 1 mL of AA were added to the mixture. The reaction was carried out under ultraviolet (UV) irradiation at room temperature for 5 h. After centrifugation and washing twice with ethanol, SiO_2_@MPSQ-AA composite particles were obtained. MPSQ-AA bowl-shaped particles were obtained by etching with 5% HF followed by centrifugation and washing twice with ethanol. Finally, bowl-shaped Janus particles were prepared using a thiol–ene click reaction as follows: MPSQ-AA bowl-shaped particles (0.5 g) were dispersed in acetonitrile (50 mL) at room temperature under mechanical stirring. Nitrogen gas was purged into the system at a flow rate of 100 mL min^−1^ to replace air. After increasing the stirring speed to 300 rpm, 1 mL of styrene was added dropwise to the mixture, and the reaction mixture was continuously stirred for 2 h under UV irradiation at room temperature. AA-MPSQ-St bowl-shaped Janus composite particles were obtained after centrifugation and washing twice with ethanol.

### 2.4. Preparation of Surfactant-Free n-Heptane-in-Water Emulsion and Oil–Water Separation Procedure

First, the emulsions were prepared at different oil-to-water volume ratios, and n-heptane dyed with oil red O was used as the oil phase. An n-heptane-in-water emulsion without surfactants was prepared by ultrasonication for 1 h. Then, 25 mg of Janus particles was added to the emulsion (5 mL) and shaken thoroughly for 60–300 s. Finally, the separation abilities of the emulsions were assessed by measuring the transmittance of the water phase.

### 2.5. Characterization

The size distribution and zeta potential of the particles in the ethanol suspension were measured using dynamic laser scattering (DLS) at the nanoparticle size and a zeta potential analyzer instrument (Nano-ZS90, Malvern Instruments, Malvin city, UK). The morphology and surface element distributions of the composite particles were characterized by scanning electron microscopy (SEM, FEI Nova NanoSEM450, Hillsboro, OR, USA, operated at 1 kV) and energy-dispersive X-ray spectroscopy (EDS, EDAX, USA). To identify the chemical constituents of the materials, Fourier transform infrared (FT-IR) spectra were recorded on a VECTOR-22 spectrometer (Bruker, Karlsruhe, Germany) using KBr pellets in the wavelength range 4000–400 cm^−1^. A digital camera and an optical microscope (OM, SMART-POL, Optec, Chongqing, China) were used for the imaging of the prepared emulsions, and the average droplet sizes were estimated using the Nano Measurer software (Nano-ZS90, Malvern Instruments, UK). Contact angles of water droplets on the particles and interfacial tension were measured using a contact angle analyzer (DAS30, KRÜSS Co., Hamburg, Germany). The transmittance of the water phase was measured using an ultraviolet–visible (UV–Vis) spectrophotometer (VARIAN CARY300, USA) at room temperature. 

## 3. Results and Discussion

### 3.1. Preparation and Characterization of Snowman-like Particles 

To identify the chemical compounds of the prepared particles, the FT-IR spectra were recorded (Figure 1). In the spectrum of SiO_2_, the absorption peak at 952 cm^−1^ was assigned to the stretching vibration of Si–OH. The stretching vibration peak caused by −SH was observed at 2554 cm^−1^ in the spectrum of SiO_2_@MPSQ. The weakening of Si–OH and the appearance of –SH indicated that the MPTMS reacted with Si–OH on the silica surface. 

Figure 2a shows the size distribution and particle morphology of the Stöber silica particles obtained by DLS and SEM, respectively. The silica particles exhibited a uniform spherical shape with an average particle size and a PDI (polydispersity index) of 243 nm and 0.059, respectively. The SiO_2_@MPSQ composite particles prepared using SiO_2_ particles showed a snowman-like morphology (Figure 2b–d). The corresponding EDS element mappings of SiO_2_@MPSQ showed the distribution of C, Si, and S elements in the samples (Figure 2c1,d1). The sulfur content of the snowman-like SiO_2_@MPSQ microspheres was 0.68 and 5.21% in the smaller and larger spheres, respectively. These results indicated that MPSQ was inhomogeneously coated on the surface of the silica template particles. It can be further inferred that the polymerization rate of MPSQ varied in different areas on the surface of the SiO_2_ template, which eventually resulted in the formation of a snowman-like morphology.

The morphology of the SiO_2_@MPSQ particles was controlled by adjusting the SiO_2_ and MPTMS ratio. The morphologies of the composite particles during the synthesis were systematically investigated by SEM. In the preparation of the composite particles, the ratios of SiO_2_ particles to MPTMS were set to 0.528, 1.060, 1.580, 1.820, and 2.110. As shown in Figure 3a,b, when the ratio of SiO_2_ particles to MPTMS was 0.528 or 1.060, the composite particles exhibited a core–shell morphology, and the thickness of the shell increased with increasing the amount of MPTMS. As the content of MPTMS continued to increase, the particle shapes gradually changed to a snowman-like morphology. Based on the above results, we suggest that the mechanism for the formation of the snowman-like particles consisted of two steps. When MPTMS was added to the aqueous suspension of SiO_2_ particles, it was hydrolyzed to produce water-soluble silanols, which rapidly condensed and grew on the surface of the silica to form a three-dimensional (3D) branched network. When the 3D network did not reach critical solubility, the composite particles formed a core–shell structure. When the critical solubility was reached, phase separation occurred on the surface of the silica particles, which resulted in the uneven growth of the MPSQ. 

In addition, we investigated the effect of the stirring speed on the morphology of the snowman-like composite particles (Figure 4). When the stirring speed was 100 rpm, the snowman-like particles showed more than one knob (Figure 4a). Uniform snowman-like particles were obtained at a stirring speed of 200 rpm (Figure 4b). However, when the stirring speed was increased to 400 rpm, irregular snowman-like particles were generated (Figure 4c). The particle morphology became non-uniform, possibly because the strong shearing caused the aggregation of colloidal particles, resulting in demulsification, and the high rotational speed could have increased the probability of collisions between the particles, thus causing an irregular morphology and even the shedding of template particles. To sum up, within a certain rotational speed range, the regularity of the particles increased with increasing the stirring speed. However, uneven snowman-like particles were obtained at a very-high stirring speed, and the particle morphology prepared at 200 rpm was the best. 

### 3.2. Characterization of Bowl-Shaped Composite Particles 

To obtain bowl-shaped particles with different internal surface areas, snowman-like SiO_2_@MPSQ particles were prepared using silica microspheres with different particle sizes, such as 243 nm, 426 nm, 550 nm, and 760 nm, as templates, and then the snowman-like particles with different particle sizes were etched with 5% HF. As shown in Appendix A, with the increase in the template particle size, the size of the snowman-like particles increased significantly. As shown in Figure 5, with the increase in template diameter, more knobs appeared on the particle surface, and the snowman-like particles showed a uniform morphology when SiO_2_ template particles with diameters of 243 or 426 nm were used. To calculate the inner surface area of the bowl-shaped particles, the small head of the snowman-like particles was assumed to be half the volume of the SiO_2_ template particles. According to the spherical surface area formula, S_sphere_ = 4πr^2^, where r is the radius of the SiO_2_ template particles, the inner surface area of the bowl-shaped particles was half of the spherical surface area after etching, that is, S_bowl_ = 2πr^2^. When the diameter of the SiO_2_ template particles increased, the opening was equal to the diameter of the template particles, and the inner surface area of the bowl-shaped particles was larger. Therefore, because of their high uniformity and large internal surface area, the snowman-like particles prepared from SiO_2_ particles with a size of 426 nm were used in the subsequent experiments.

### 3.3. Characterization of Janus Composite Particles

The SEM images of the snowman-like SiO_2_@MPSQ-AA, bowl-shaped MPSQ-AA, and bowl-shaped Janus composite particles (AA-MPSQ-St) prepared from SiO_2_ templates with a diameter of 426 nm are shown in Figure 6. The snowman-like SiO_2_@MPSQ-AA particles after the click reaction with acrylic acid showed no changes in their particle shape (Figure 6a). Figure 6b shows the bowl-shaped particles after the selective etching by HF with the removed SiO_2_ template particles, resulting in a regular bowl-shaped morphology. In addition, Figure 6c shows the bowl-shaped Janus particles after the click reaction with styrene. The particle morphology did not change significantly, and the prepared bowl-shaped particles still exhibited good uniformity.

The chemical structures of the materials during preparation were determined using FT-IR spectroscopy, as shown in Figure 1. In the spectrum of SiO_2_@MPSQ-AA, the peaks at 1730 cm^−1^ were attributed to the stretching vibration of the carbonyl group (C=O) in AA. In the spectrum of the bowl-shaped MPSQ-AA particles, the peak corresponding to the –SH groups reappeared at 2554 cm^−1^ after etching, indicating that the –SH functionalities on the inner surface of the particles were exposed, enabling the next reaction step. For the bowl-shaped Janus composite particles, the absorption bands at 1600 and 1492 cm^−1^ were associated with the stretching vibrations of the benzene ring skeleton of styrene, and the peaks at 756 and 698 cm^−1^ originated from the flexural vibrations of a single substituted benzene ring. The peak at 3025 cm^−1^ was attributed to the stretching vibrations of the unsaturated C−H bonds in the benzene rings. The above FT-IR results proved that the MPSQ-AA-St bowl-shaped Janus composite particles were successfully obtained.

The static contact angles of water on the different particles were measured in air to investigate the surface wettability of the particles. The results are shown in Figure 7a. The surface of the silica particles prepared by the Stöber method had hydroxyl groups, which provided the particles with an excellent hydrophilicity. For the snowman-like particles, the surface of hydrophilic SiO_2_ particles was coated with hydrophobic polysiloxane, resulting in an increase in the contact angle from 10.2° to 152.4° due to the superhydrophobic particle surface. The contact angle decreased again to 20.6° after the snowman-like particles were modified with acrylic acid containing hydrophilic carboxyl groups. After the etching with 5% HF, the contact angle changed to 124.8° because of the exposure of the hydrophobic polysiloxane on the inner surface of the bowl-shaped particles. Owing to the successful modification with styrene, the contact angle of the Janus particles further increased to 156.9°. To further confirm the hydrophobic, hydro-, or amphiphilic behaviors of the particles, they were added to an oil–water emulsion. As shown in Appendix A, the hydrophilic snowman-like particles grafted with acrylic acid precipitated at the bottom of the water phase, the hydrophobic snowman-like particles before grafting floated on the surface of the oil phase, and the amphiphilic Janus particles were at the interface of the two phases. The zeta potentials of the three particles are shown in Figure 7b. The absolute value of the zeta potential decreased with the proceeding modification. These results also verified that styrene was successfully grafted onto the inner surface of the bowl-shaped particles.

### 3.4. Oil–Water Separation of Amphiphilic Janus Particles

The separation of oil droplets from water using the bowl-shaped Janus particles was also investigated. As shown in Figure 8a, when the bowl-shaped Janus particles were added to the n-heptane–water emulsion and shaken for 300 s, the stable emulsion formed flakes, and the color of the solution became lighter. In the absence of Janus particles, the appearance of the emulsion did not significantly change, even after shaking for 300 s. It became slightly clearer after 12 h, indicating the destabilization of the emulsion.

The effect of the shaking time on the separation of oil and water was studied using a UV spectrophotometer. The effect of the shaking time on the oil–water separation was studied by measuring the water phase transmittance. As shown in Figure 8c, the water-phase transmittance increased with increasing the shaking time. After shaking for 60, 120, and 180 s, the oil–water emulsions were not completely separated, which only occurred when the shaking time was increased to 240 or 300 s, indicating that an effective oil–water separation occurred within 300 s. Simultaneously, the separation efficiency increased when higher amounts of Janus particles were added (Figure 8d). The separation performance of the Janus particles in the oil–water mixtures with different n-heptane contents was also studied. The results showed that the separation was effective even at a low n-heptane volume fraction of 0.5% (Appendix A). In addition to n-heptane, other types of oils, such as cyclohexane and toluene, could also be effectively separated from water using our Janus particles (Appendix A). These results indicated that the Janus particles could be used to achieve an effective and quick separation of oil droplets from oil–water mixtures. 

To understand the mechanism of the fast and efficient oil droplet separation process, a comprehensive investigation of the adsorption and agglomeration of oil droplets mediated by Janus particles was conducted. Before the addition of the bowl-shaped Janus particles, the n-heptane–water emulsion was observed using a digital camera and a light microscope. The images below show that the mixture consisted of oil droplets with diameters less than 5 µm (Figure 9a). When the bowl-shaped Janus particles were added to the emulsion and shaken, the number of oil droplets was significantly reduced (Figure 9b). Under constant shaking, the emulsion rapidly formed large oil droplets in the upper layer induced by bowl-shaped Janus particles. In addition, the bowl-shaped Janus particles acted as a surfactant that bound to the interface of the large oil droplets.

The classical nucleation theory can be used to understand the coalescence process of oil droplets [5]. When two oil droplets approach each other to merge into a bigger oil droplet, they form a bridge with a radius *R* (Figure 9d). The free energy of the bridge can be described as follows. The line tension *γ_L_* of the bridge, which results from the strong bending of the interface, can be written as
(1)$$F_{\mathrm{line}\;\mathrm{tension}}=2\pi R\gamma_{L}$$
where *R* is the bridge radius. Moreover, the decrease in the interface free energy can be described using Equation (2).
(2)$$F_{\mathrm{surface}\;\mathrm{tension}}=-2\pi R^{2}\gamma$$

The final free energy of the bridge was obtained by combining these two contributions:(3)$$F\left(R\right)=2\pi R\gamma_{L}-2\pi R^{2}$$

A typical plot of the free energy versus the bridge radius is shown in Figure 9e [5]. The line tension is the main contributor for a small radius, whereas, for a large radius, the contribution of surface tension is dominant. The radius of the bridge should be sufficiently large ($R>R^{*}=\gamma_{L}/2\gamma$), and the energy barrier ($\Delta F^{*}=\pi \gamma_{L}^{2}/2\gamma$) should be overcome to enable coalescence of the droplets. 

The pendant droplet method was used to measure the interfacial tension of the bowl-shaped Janus particles at the oil–water interface. For this, an aqueous suspension of Janus particles with a concentration of 15 mg mL^−1^ was dropped into the n-heptane phase. As shown in Figure 10a, the tension at the water–n-heptane interface decreased during the formation of the water droplets containing Janus particles, eventually approaching a new equilibrium value. 

We described the interfacial activity of Janus particles by calculating their absorption energy (Δ*G*). The free energy required for the desorption of bowl-shaped Janus particles from the oil–water interface to the water phase is defined as $\Delta G=F_{\mathrm{water}}-F_{\mathrm{surface}}$. 

We assumed that the bowl-shaped Janus particles stayed at the contact line of the oil–water interface and the St-AA boundary. For simplicity, the oil–water interface was considered flat, and *S* was defined as the oleophilic inner surface area of the bowl-shaped Janus particles. For the bowl-shaped Janus particle, the free energies *F*_water_ and *F*_surface_ can be calculated as follows: (4)$$F_{\mathrm{water}}=2\pi r^{2}\gamma_{\mathrm{water}-\mathrm{AA}}+S\gamma_{\mathrm{water}-\mathrm{St}}+\pi r^{2}\gamma_{\mathrm{water}-\mathrm{oil}}$$
(5)$$F_{\mathrm{surface}}=2\pi r^{2}\gamma_{\mathrm{water}-\mathrm{AA}}+S\gamma_{\mathrm{oil}-\mathrm{St}}$$

Based on Equations (4) and (5), the adsorption energy of a bowl-shaped Janus particle can be described using Equation (6):(6)$$\Delta G_{\mathrm{bowl}-\mathrm{shaped}}=S\left(\gamma_{\mathrm{water}-\mathrm{St}}-\gamma_{oil-St}\right)+\pi r^{2}\gamma_{\mathrm{water}-\mathrm{oil}}$$

According to the above formula, Δ*G* increases as the hydrophobic inner surface area of bowl-shaped Janus particles increases, indicating that Janus particles with larger oleophilic internal surface areas can adsorb oil droplets more easily. 

Compared with other reports in the literature [5], the prepared particles could carry out a high-efficiency separation of oil and water in a similar time, and the separation efficiency was 98% or above. At the same time, the morphology of the Janus particles used for the oil–water separation was more regular, and the mass of the particles used was smaller. The recyclability of the particles needs to be explored in the future research.

## 4. Conclusions

We designed a method for preparing snowman-like particles based on SiO_2_ particles using the sol–gel approach. The snowman-like particle morphologies were controlled by varying the ratios of SiO_2_ template particles to MPTMS monomers. At low ratios of 0.528 and 1.060, core–shell composite particles were formed. When the molar ratio increased to 2.110, the morphology changed from a core–shell morphology to snowman-like particles. In addition, when the rotational speed was set at 200 rpm and SiO_2_ particles with diameters of 243 and 426 nm were used as templates, the morphology of the snowman-like composite particles was the most regular and uniform. Further, we used snowman-like particles to prepare bowl-shaped Janus particles by a thiol–ene click reaction and selective etching. The contact angle measurements confirmed the amphiphilic nature of the prepared bowl-shaped Janus particles, which could separate various types of oil droplets from water with a separation efficiency of more than 98% within 300 s. This article provides a simple method for preparing Janus particles, which broadens the application range of Janus particles. At the same time, the Janus particles designed and prepared in this paper provide a simple new idea for designing anisotropic particles. We expect our materials to have good application prospects for oil separation from industrial wastewater and water purification.

## Data Availability

The data presented in this study are available on request from the corresponding author.

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
