# Peer review of "Amphiphilic Bowl-Shaped Janus Particles Prepared via Thiol–Ene Click Reaction for Effective Oil–Water Separation"

_nanomaterials, 2023, doi:10.3390/nano13030455_

Round 1

Reviewer 1 Report

This paper is on preparing a special nanostructure with interesting morphological features. The features include snowman-like and Janus type structures. Then, the oil-water separation studies were nicely presented to show the application of the nanostructure. The quality of the paper is good for publishing.

Minor touches:

1. In reference section, item number 2 may add [J] at the end of the article title to keep the format consistency. In addition, some article titles are with lower case letters (in [1], [2], [3] etc.) some are fully capitalized (in [4] and [5]). It is suggested keep the same format.

2. In Results and Discussion Section, on line 165, "energy-dispersive X-ray (EDS) spectra" should be "X-ray diffraction energy-dispersive spectra (EDS)"

Reviewer 2 Report

nanomaterials-2150447-peer-review-v1

After carefully evaluation. I am pleased to send you some comments. Please consider these suggestions as listed below to prepare the article again.  

  1. The title seems ok but its better if author correlate according to journal.
  2. The abstract seems to be good. Please add one more introductory line of your objective in beginning of abstract.
  3. Key words ???? please give atleast 4-5 most suitable keywords.
  4. Research gap should be delivered on more clear way with directed necessity for the future research work.
  5. Introduction section must be written on more quality way, i.e., more up-to-date references addressed.
  6. The novelty of the work must be clearly addressed and discussed, compare previous research with existing research findings and highlight novelty.
  7. What is the main challenge?
  8. Please check the abbreviations of words throughout the article. All should be consistent.
  9. What is problem statement?
  10. The main objective of the work must be written on the more clear and more concise way at the end of introduction section.
  11. Do not use lumpy reference. Such as 8-19, 20-24 etc. Its very weired. The author should use maximum 2-3 reference at one place. Please revise your paper accordingly since some issue occurs on several spots in the paper.
  12. There are several irrelevant reference. Please remove it.
  13. Please provide space between number and units. Please revise your paper accordingly since some issue occurs on several spots in the paper.
  14. Pleas remove reference 1 to 3 and simple cite this single reference here at Line 30. Carbon Nanocomposite. In Environmental Remediation through Carbon Based Nano Composites; Springer: Berlin/Heidelberg, Germany, 2021; pp. 1–18.
  15. Similar remove 8-19 and simply cite this single article here- Umar, K.; Yaqoob, A.A.; Ibrahim, M.N.M.; Parveen, T.; Safian, M.T. Environmental applications of smart polymer composites. Smart Polym. Nanocompos. Biomed. Environ. Appl. 2020, 15, 295–320.
  16. The methodology part is very weird. Please revise it carefully and describe the procedure clearly.
  17. Please highlights the peaks in figure 1. Also it’s not well scientifically explained.
  18. The entire result section there is no scientific support for your result why?
  19. The results does not have any scientific discussion.
  20. Why author explain the results in a very general way.
  21. Please include all chemical/instrumentation brand name and other important specification.
  22. Please provide high quality image for figure 10a.
  23. Regarding the replications, authors confirmed that replications of experiment were carried out. However, these results are not shown in the manuscript, how many replicated were carried out by experiment? Results seem to be related to a unique experiment. Please, clarify whether the results of this document are from a single experiment or from an average resulting from replications. If replicated were carried out, the use of average data is required as well as the standard deviation in the results and figures shown throughout the manuscript. In case of showing only one replicate explain why only one is shown and include the standard deviations.
  24. Please add a comparative discussion section.
  25. Conclusion and Future perspectives should be revised carefully. Conclusion section is missing some perspective related to the future research work, quantify main research findings, and highlight relevance of the work with respect to the field aspect.
  26. To avoid grammar and linguistic mistakes, MAJOR level English language should be thoroughly checked. Please revise your paper accordingly since several language issue occurs on several spots in the paper.
  27. Reference formatting need carefully revision. All must be consistent in one format. Please follow the journal guidelines.  The table and figure formatting also need carefully revision.

The current format of the presentation indicates that the author did not make any genuine attempts to prepare it. The formatting is strange, and they specifically neglected the result and comment sections to explain in a suitable manner.

Decision = Encourage you to resubmit after careful revision as a new version. It was tough for me to read and go through, but I was able to make a few remarks. Please put forth your best efforts and resubmit as a new submission. Sorry I cannot recommend directly as Nanomaterial is very reputed journal.

Reviewer 3 Report

The paper is well written and presents interesting results of the modified magnetic particles. 

The characterization was carried out but the display of the obtained information is not good. Most of the figures have low resolution (poor) and are not readable..  The values for zeta potential and size measurements need to include the error value associated.  These particles may have been studied at different pH (pH Titration) to evaluate the effect of pH on the zeta potential and the size (agglomeration)  SEM image has very low resolution, and it seems to show clustering / agglomeration, which is not beneficial for the aimed task. The authors must comment on this.  In addition, TEM analysis is highly suggested.  IR spectra of the functionalized particles may be another way to determine the efficiency of the functionalization stage.  I suggest major revisions.   

Round 2

Reviewer 2 Report

Dear Authors
I have reviewed again the manuscript and I think that it is ready for publication. Thank you for considering my suggestions.